# Time trends in and factors associated with repeat adolescent birth in Uganda: Analysis of six demographic and health surveys

Dinah Amongin[1,2]*, Annettee Nakimuli[1], Claudia Hanson[3,4], Mary Nakafeero[5], Frank Kaharuza[6], Lynn Atuyambe[6], Lenka Benova[7,8]

1 Department of Obstetrics and Gynaecology School of Medicine, Makerere University College of Health Sciences, Kampala, Uganda, 2 Department of Health Policy Planning and Management, Makerere University School of Public Health, Kampala, Uganda, 3 Department of Disease Control, London School of Hygiene & Tropical Medicine, London, England, 4 Department of Public Health Sciences, Karolinska Institutet, Stockholm, Sweden, 5 Department of Epidemiology and Biostatistics, Makerere University School of Public Health, Kampala, Uganda, 6 Department of Community Health and Behavioral Sciences, Makerere University School of Public Health, Kampala, Uganda, 7 Faculty of Epidemiology and Population Health, London School of Hygiene and Tropical Medicine, London, England, 8 Department of Public Health, Institute of Tropical Medicine, University of Antwerp, Antwerp, Belgium

* amongdinah2003@yahoo.com

**Data Availability Statement:** All data are available under the DHS program. URLs: https:// dhsprogram.com/data/dataset_admin/login_main.

## Abstract

### Background

Information on repeat adolescent birth remains scarce in sub-Sahara Africa. We investigated the prevalence and time trends in repeat adolescent birth in Uganda, and associated factors.

### Methods

We analyzed Uganda Demographic and Health Survey data of women age 20–24 years collected on 6 surveys (1988/89-2016) to estimate repeat adolescent birth (first live birth <18 years of age followed by another live birth(s) <20 years). Further, we estimated the wantedness of the second order birth and the prevalence of short birth intervals birth (<13 months) between the first and second such birth. On the 2016 survey, we examined factors associated with repeat adolescent birth using bivariate and multivariate modified Poisson regression.

### Results

At the 1988/89 survey, 58.9% of women with first birth <18 years reported a repeat adolescent birth. This percentage increased to 66.8% in 2006 (+7.9 percentage points [pp], p = 0.010) and thereafter declined to 55.6% by 2016 (-11.2 pp, p<0.001), nevertheless, no change occurred between 1988/89 and 2016 (-3.3pp, p = 0.251). Among women with repeat adolescent births, the mean number of live births by exact age 20 years (2.2 births) and prevalence of short birth intervals (3.5% in 1988/89, 5.4% in 2016) (+1.9pp, p = 0.245) did not change. Increasingly more women with repeat adolescent births preferred to have had

cfm?CFID=82350&CFTOKEN=c579510c80a1f091-57ED4B37-B9E4-6030-73D516CE4DCAA73E.

**Funding:** This work was supported through the Developing Excellence in Leadership, Training and Science (DELTAS) Africa Initiative grant # DEL-15-011 to THRiVE-2. The DELTAS Africa Initiative is an independent funding scheme of the African Academy of Sciences (AAS)'s Alliance for Accelerating Excellence in Science in Africa (AESA) and supported by the New Partnership for Africa's Development Planning and Coordinating Agency (NEPAD Agency) with funding from the Wellcome Trust grant # 107742/Z/15/Z and the UK government. The views expressed in this publication are those of the author(s) and not necessarily those of AAS, NEPAD Agency, Wellcome Trust or the UK government. DA is the only author who received this award to conduct the study. URLs to sponsors website: https://thrive.or.ug/ https://www.aasciences.africa https://au.int/en/NEPAD https://wellcome.ac.uk/ The funders had no role in study design, data collection and analysis, decision to publish, or preparation of the manuscript.

**Competing interests:** The authors have declared that no competing interests exist.

the second child later, 22.5% in 1995 and 43.1% in 2016 (+20.6pp, p = <0.001). On the 2016 survey, women from poorer households and those of younger age at first birth were significantly more likely to report repeat adolescent birth.

## Conclusion

Following a first birth <18 years, more than half of the women report a repeat adolescent birth (<20 years), with no decline observed in 30 years. Increasingly more women wanted the second adolescent pregnancy later, highlighting the need to support adolescents with improved family planning services at each contact.

## Introduction

Globally, adolescent childbearing remains a major public health concern most especially in the low- and middle-income countries (LMICs). Approximately 1 in 8 of the 140 million births annually occurs to adolescent women with 95% of these occurring in LMIC, and 23% in sub-Sahara Africa [1–4]. Uganda has a high adolescent childbearing rate with estimates at 25% of 15–19 year old having begun childbearing [5]. These levels have remained high in the last 15 years despite a decline in the age specific fertility rate for Uganda among women 15–19 years, 195 births per 1,000 women per year in the 1990 Uganda Demographic and Health Survey (UDHS) to 132 births per 1,000 women per year at the 2016 survey.

Adolescent childbearing lays a foundation for disadvantages in the areas of health, social, and economic outcomes both in the short term and long term [3, 4, 6–9]. Adolescent women and their babies are at higher risk of experiencing poor health outcomes such as obstetric fistula, sepsis, stillbirths, preterm births, birth asphyxia, poor child survival and mental disorders. Socially, adolescent women often face violence from family and community, discrimination and stigma with a high risk of economic disadvantages compounded by premature cessation of schooling and early marriage [10–12]. The younger the adolescent mother, the more vulnerable she is both socio-economically and medically to poor outcomes, including repeat pregnancies [13–15]. Not seldomly, adolescent pregnancies are result of sexual and gender based violence [16–19]. Experiencing another birth before 20 years of age (= repeat adolescent childbirth) may therefore push the adolescent woman and her offspring into worse outcomes than what she experienced following the first birth. Repeat pregnancy in adolescence is more common in settings of high poverty, low educational attainment or its discontinuation, early union or being in a union, none use of long acting reversible contraceptives and previous abortion or non-live birth, among others [20–23].

Information on the extent of repeat adolescent birth, even though it potentially constitutes a large portion of adolescent fertility, is scarce. Second and higher order births among <20 year olds are in some countries a substantial percentage of all such births [24]. Existing studies examined repeat adolescent pregnancies mainly in high income countries with few focusing on repeat adolescent birth, as an end point. These studies, mainly prospective cohorts using health facility samples and systematic reviews, estimated prevalence of repeat pregnancies among adolescents at approximately 17% [21, 22, 25–28]. We identified one study from the Philippines, one from Thailand-Myanmar boarder and three studies from sub-Saharan Africa: Tanzania, South Africa and Uganda [23, 29–32]. The Uganda study, estimated rapid repeat pregnancy (within 12 months) among women 15–22 years at 37% and 74% within 24 months [23]. Whether these repeat pregnancies were wanted then or later, was not determined.

Wantedness of pregnancy, though it may not translate into the actual births, provides information on unwanted pregnancies and desired birth intervals [33, 34]. Birth intervals between the first and second birth among women in LMICs, 15–49 years, have lengthened overtime in tandem with fertility decline between 1965 and 2014 [35]. The short birth intervals <24 months declined as the interval between first and second birth increased. Information on initiation of childbearing in adolescence has received some attention in Uganda [23, 36, 37]. Studies have reported a decline in first adolescent childbirth in Uganda [38, 39], but little is known about repeat adolescent birth.

We estimated the levels and time trends of repeat adolescent births (another birth below 20 years) in Uganda using all available six rounds of the UDHS data from 1988/89 to 2016 among women with first birth <18 years of age with a view to inform programming for and delivery of adolescent health care services. We also estimated the percentage of all women age 20–24 years who reported a repeat adolescent birth. The birth intervals between the first and second order adolescent birth and wantedness of the second order birth at that time point were also assessed. Last, using the 2016 UDHS data, we examined factors associated with repeat adolescent birth.

## Methods

### Data source, population and definitions

We analysed data from all six UDHS rounds (1988/89, 1995, 2000/01, 2006, 2011, and 2016). The DHS are nationally representative cross-sectional surveys that collect information on population, including on maternal and child health. In these surveys, two-stage cluster sampling was conducted with representation of all the geographic regions of a country. All the data was from women's self-report and was collected by trained data collectors. The interviewer-administered questionnaires used were translated into the local languages and pre-tested prior to data collection. Due to security concerns, 20% of the country was not assessed in 1988/89 and 5% in 2000/01 surveys.

The analysis sample was women aged 20–24 years at the time of each survey. This age category was chosen because of completion of the age at risk of adolescent birth, based on the World Health Organization definition of adolescence (10-19years of age) [40].

Our main outcome was repeat adolescent birth, defined as first live birth <18 years of age followed by another live birth(s) <20 years among 1) women 20–24 years old with first live birth <18 years and 2) among all women 20–24 years. We used live birth as an outcome (rather than pregnancy) as this information is consistently available in the DHS.

Birth intervals were calculated between the first and second order live birth among women with repeat adolescent birth. We categorized the intervals into three groups; <13 months (short birth interval), 13–24 months and above 24 months. All the women had completed the period of observation of 12 months, according to our definitions. Among a subset of women for whom the second birth had occurred within the 5 years preceding the survey (i.e. during the survey recall period during for which this question was asked), we assessed whether the women wanted the pregnancy then, later or wanted no more (= wantedness). Information on wantedness of that birth was available for all surveys except 1988/89.

The 2016 survey data set was chosen for analyzing the factors associated with repeat adolescent birth as this was the most recent survey that would provide the most current information on these factors. The risk factors explored in multivariable analysis were socio-demographic position (region, residence, religion, and household wealth quintile) at the time of the survey, and sexual/reproductive health predictors—age at first sex and age at first birth. Regions were categorized into four based on the categorization at the 2000/01 survey. Central region

contained Kampala, Central 1 and Central 2. Northern region included Lango, Acholi and West Nile sub-regions. Eastern region was composed of Teso, Karamoja, Bugisu, Bukedi, and Busoga sub-regions while Western region was composed of Bunyoro, Tooro, Ankole and Kigezi sub-regions. Residence and household wealth quintile were maintained as captured in the survey data set. Religion was re-categorized into 4 categories; Anglican, Catholic, Muslim and Other. Catholic and Muslim religions were not re-categorized. Anglican religion included Anglican and Pentecostal/born again/evangelical. "Other" religion category included all the remaining groups; Seventh day Adventist, orthodox, Baptist, traditional, no religion, other, and Jehovah's Witness. Due to possibility of reverse causality, the following potential factors were not included in the analysis: education level at time of survey, occupation at time of survey, and marital status at survey. Previous studies have indicated that education level, occupation and marital status are predictors of first birth [41] and therefore, by the time of the second order birth, reverse causality may be seen. The final variables included were chosen based on availability of this information in the DHS data and predictors from previous studies. Contraception use was excluded as there was no information on its use at the point of the repeat adolescent pregnancy. The DHS collects information about contraception use at the point of the survey.

## Analysis

Analysis was performed using STATA version 12.0, StataCorp LP, Texas. Sample weights were applied for all analysis to have a sample that is representative of the whole population and reduce sampling bias [42, 43]. The issue of multiple (twins or triplets etc) births was factored in when categorizing into repeat and no repeat adolescent births and calculating the birth intervals by counting each delivery event as one birth, irrespective of multiplicity.

We calculated the absolute percentage point (pp) differences in outcomes between survey points and used the two-sample test of proportions to estimate the p-value of differences. Descriptive statistics for the characteristics using proportions for categorical variables and means with standard deviations for continuous variables were presented. Birth intervals between the first and second birth and wantedness of the second order live birth were presented as proportions.

In determining the factors associated with repeat adolescent birth on the 2016 survey, we calculated the crude and adjusted prevalence risk ratios. The modified Poisson regression was used for this analysis and it was chosen over the logistic regression in this cross sectional study, with a binary outcome, so as to avoid odds ratios over estimating the prevalence ratios in our scenario where the likelihood of the outcome was high, above 10% [44]. All factors were included in the final models irrespective of whether they were significant at crude analysis or not. The total sample of women with first birth <18 years was 1084 and there were no missing values in variables to calculate the outcome. In calculation of the mean age at first sex, 47 women were classified as inconsistent (age at first sex indicated as having occurred after childbirth) and were excluded from this analysis.

## Ethics

The School of Medicine Research Ethics Committee (SOMREC) Makerere University and the Uganda National Council for Science and Technology (UNCST) gave ethical approval for the study. Permission to access and use the data sets was sought from the DHS program that collects data after obtaining approvals from the Government of Uganda and informed consent from respondents during the survey.

## Results

### The prevalence and time trends in repeat adolescent birth

In Table 1, the data in the last row "among women 20–24 years with first birth <18 years" is a sub-set sample of those with a first birth before 18 years (the two rows directly above). Among the sample of all women age 20–24 years, those reporting first birth <18 years reduced from 41.7% (411/985) at the 1988/89 to 28.4% (1084/3822) at 2016 survey. The percentage who reported repeat adolescent birth following a first birth <18 years of age was 24.6% (242/985) at the 1988/89 survey, increased to 26.6% (399/1504) at the 2000/01 and thereafter declined to 15.8% (603/3822) at the 2016 survey (Table 1). Overall, the percentage of the women 20–24 years reporting repeat adolescent birth declined between 1988/89 and 2016 (-8.8 percentage points [pp], p<0.001) (S1 Table).

Among women 20–24 years with first birth <18 years, 58.9% (242/411) of the women in 1988/89 survey reported a repeat adolescent birth compared to 55.6% (603/1084) at the 2016 (Table 1 and Fig 1). There was no change in repeat adolescent birth in the initial 15 years (+4.4pp, p = 0.154) compared to the later 15 years, 2006–2016 (-11.2pp, p<0.001). Overall, the percentage of women with first birth <18 years reporting repeat adolescent birth in 1988/89 and 2016 were similar- a -3.3pp difference (p = 0.251).

Among all the women 20–24 years, the mean number of births by exact age 20 years declined from 0.95 to 0.69 at the 1988/89 and 2016 surveys, respectively (Table 2). The decline in mean and median number of live births started after the 2000/01 survey. Among the women with repeat adolescent birth following first birth <18 years, the mean (2.2) and median (2) number of births by exact age 20 years did not change across the surveys. Further, the percentage of those with repeat adolescent birth reporting 3 or more children by exact age 20 years did not change in the 30 years, 19.8% in 1988/89 and 20.2% in 2016.

### Birth interval between first and second birth order

The percentage of women with repeat adolescent birth reporting a birth interval of <13 months was 3.5% (95% CI 1.6–7.7) at the 1988/89 survey with highest proportion being 6.7% at the 2000/01 survey. During the entire period of observation, there was no change in the extent of birth intervals <13 months, with a +1.9pp difference (p = 0.245) in this birth interval between 1988/89 and 2016 surveys. Women reporting a birth interval between 13–24 months remained at just over 40% (Table 3).

**Table 1. Prevalence and time trends in repeat adolescent birth among Uganda women age 20–24 years (% and 95% CI); all surveys.**

| Survey | 1988/89 UDHS | 1995 UDHS | 2000/01 UDHS | 2006 UDHS | 2011 UDHS | 2016 UDHS |
|---|---|---|---|---|---|---|
| **Among all women 20–24 years (column %, 95% CI)** | | | | | | |
| | N = 985 | N = 1555 | N = 1504 | N = 1710 | N = 1629 | N = 3822 |
| **No birth <18** | 58.3 | 60.9 | 58.1 | 64.8 | 67.0 | 71.6 |
| | (54.5–62.0) | (57.7–64.0) | (54.6–61.4) | (62.1–67.3) | (63.9–69.9) | (69.8–73.4) |
| **1st birth <18, No repeat birth <20** | 17.1 | 14.8 | 15.4 | 11.7 | 12.6 | 12.6 |
| | (14.6–19.9) | (12.8–16.9) | (13.4–17.6) | (10.2–13.3) | (10.8–14.8) | (11.4–13.9) |
| **1st birth <18, Repeat birth <20** | 24.6 | 24.3 | 26.6 | 23.5 | 20.4 | 15.8 |
| | (21.5–28.0) | (21.4–27.5) | (23.7–29.7) | (21.3–26.0) | (17.9–23.2) | (14.4–17.2) |
| **Among women 20–24 years with first birth <18 years (%, 95%CI)** | | | | | | |
| | N = 411 | N = 608 | N = 631 | N = 603 | N = 538 | N = 1084 |
| **Repeat birth <20 years** | 58.9 | 62.2 | 63.3 | 66.8 | 61.7 | 55.6 |
| | (53.5–64.2) | (57.1–67.1) | (58.8–67.6) | (62.8–70.6) | (56.3–66.8) | (52.1–59.1) |

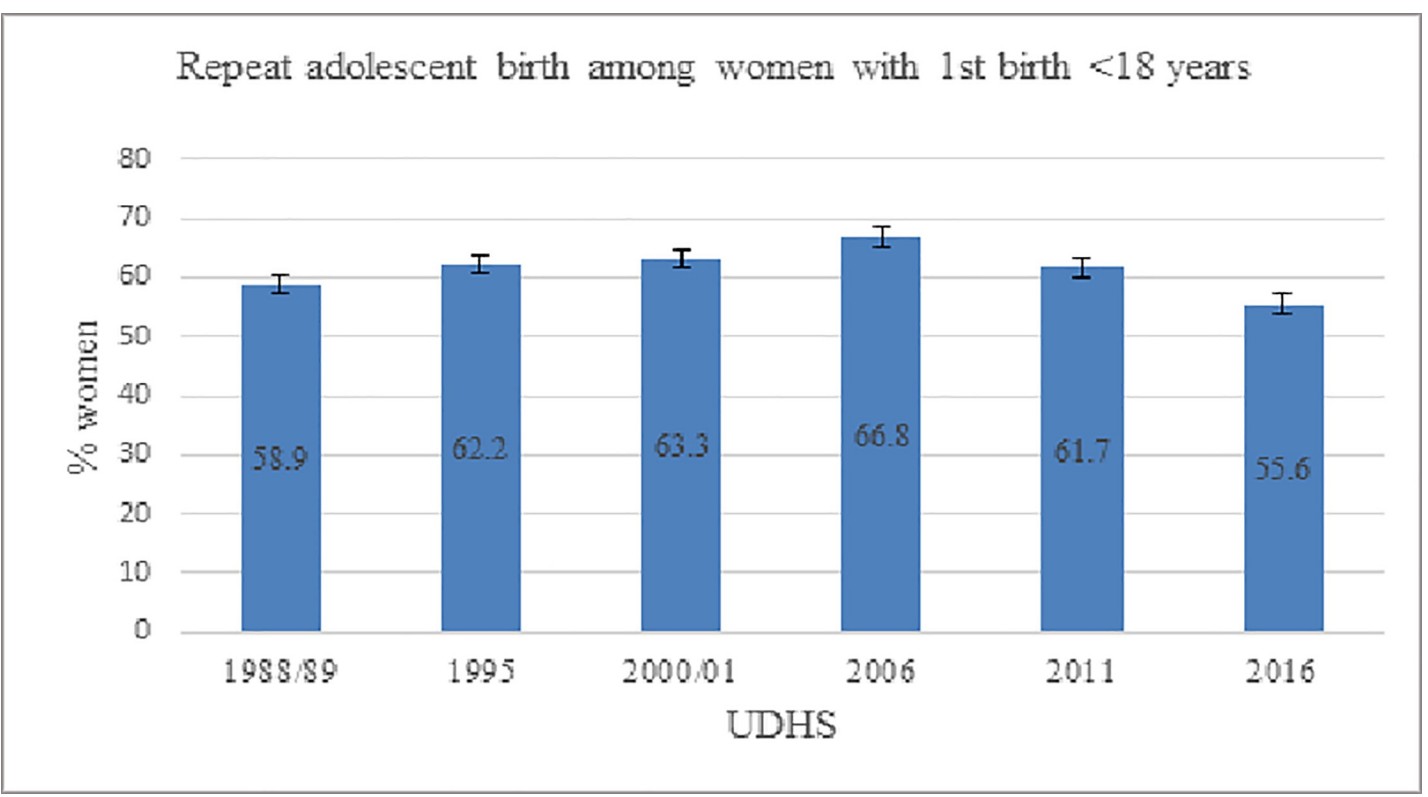

**Fig 1. Percent of women age 20–24 years with first childbirth <18 years reporting repeat adolescent birth in Uganda, by year of survey.**

### Wantedness of second order birth

We analysed women with repeat adolescent birth in whom the second live birth occurred in the 5 years preceding the survey and therefore had data on pregnancy wantedness. The percentage of women reporting having wanted that pregnancy to come later increased from 22.5% (95% CI, 15.8–31.1) at the 1995 survey to 43.1% (95% CI, 37.4–49.0) at the 2016 (+20.6pp difference, p <0.001) (Table 4).

### Factors associated with repeat adolescent birth

We analysed factors associated with repeat adolescent birth on the 2016 survey. A total of 1084 women age 20–24 years had a first birth <18 years (Table 5). In the crude associations of the various factors with repeat adolescent birth, rural residents were more likely than urban residents to have had a repeat adolescent birth (crude prevalence ratio [PR] 1.31, 95% CI = 1.08–1.60). Women from Eastern and Western regions were more likely to report the outcome compared to those from central region (PR 1.38, 95% CI = 1.13–1.69 and 1.25, 95% CI = 1.01–1.54, respectively). Each additional year decrease in age at first sex and age at first birth was associated with increased likelihood of reporting a repeat adolescent birth (PR 0.84, 95% CI = 0.81–0.88 and 0.76, 95% CI = 0.74–0.79 respectively). Women from the poorest wealth quintiles were more likely to report the outcome.

In adjusted analysis, two factors were found to be significantly associated with reporting a repeat adolescent birth: household wealth quintile and age at first birth. Women in the richer (adjusted prevalence ratio [aPR] 0.81, 95%CI 0.67–0.98) and richest (aPR 0.64, 95% CI = 0.48–0.84) household wealth quintiles were less likely to report a repeat adolescent birth compared

**Table 2. Mean and median number of live births by age 20 years (<20 years).**

| Survey | 1988/89 UDHS | 1995 UDHS | 2000/01 UDHS | 2006 UDHS | 2011 UDHS | 2016 UDHS |
|---|---|---|---|---|---|---|
| **Among all women 20–24** | | | | | | |
| **Mean no. live births (SD)** | 0.95 | 0.93 | 0.97 | 0.89 | 0.78 | 0.69 |
| | (0.9) | (0.9) | (0.9) | (0.9) | (0.9) | (0.8) |
| **Median no. live births (IQR)** | 1 | 1 | 1 | 1 | 1 | 0 |
| | (0–2) | (0–2) | (0–2) | (0–1) | (0–1) | (0–1) |
| **Among women 20–24 with first birth <18 years** | | | | | | |
| **Mean no. live births (SD)** | 1.73 | 1.79 | 1.76 | 1.85 | 1.77 | 1.69 |
| | (0.7) | (0.8) | (0.7) | (0.7) | (0.7) | (0.7) |
| **Median no. live births (IQR)** | 2 | 2 | 2 | 2 | 2 | 2 |
| | (1–2) | (1–2) | (1–2) | (1–2) | (1–2) | (1–2) |
| **Among women 20–24 with first birth <18 years and repeat adolescent birth <20 years** | | | | | | |
| **Mean no. live births (SD)** | 2.23 | 2.27 | 2.20 | 2.27 | 2.24 | 2.24 |
| | (0.5) | (0.5) | (0.5) | (0.7) | (0.5) | (0.5) |
| **Median no. live births (IQR)** | 2 | 2 | 2 | 2 | 2 | 2 |
| | (2–2) | (1–2) | (2–2) | (2–2) | (2–2) | (2–2) |
| **% of women 20–24 with first birth <18 years and repeat adolescent birth <20 years reporting 3 and more births by exact age 20 years** | | | | | | |
| **3 and more children** | 19.8 | 22.7 | 18.7 | 22.8 | 22.1 | 20.2 |
| | (14.3–26.7) | (18.3–27.8) | (14.4–24.0) | (18.5–27.7) | (17.0–28.2) | (16.8–24.0) |

*SD–standard deviation, IQR -Interquartile range

to those in the lower three poorest quintiles. Each additional year increase in age at first birth was associated with a 23% lower likelihood of reporting repeat birth (p <0.001).

## Discussion

### Prevalence and time trends of repeat adolescent birth

Our study found that approximately half of women 20–24 years reporting first birth <18 years of age had another birth at each of the six UDHS surveys. There was no significant decline in the prevalence of repeat adolescent birth over the 30-year period of observation, including the mean number of births by exact age 20 years among these women. However, in the entire sample of all women age 20–24 years, we found decline in repeat adolescent births as a result of an overall decline in women reporting first birth <18 years. The wantedness of the second order birth declined in this period with—more women report having wanted to delay the second pregnancy.

**Table 3. Birth intervals between first and second live birth among women age 20–24 years with repeat adolescent birth following first birth <18 years, all surveys (column % and 95%CI).**

| Survey | 1988/89 | 1995 | 2000/01 | 2006 | 2011 | 2016 |
|---|---|---|---|---|---|---|
| **Birth interval category** | **N = 242** | **N = 378** | **N = 399** | **N = 403** | **N = 327** | **N = 603** |
| **<13 months** | 3.5 | 3.4 | 6.7 | 4.4 | 4.0 | 5.4 |
| | (1.6–7.7) | (1.9–5.8) | (4.0–10.8) | (2.5–7.6) | (2.2–7.3) | (3.8–7.5) |
| **13–24 months** | 44.3 | 42.2 | 48.0 | 50.0 | 41.8 | 42.7 |
| | (37.9–50.8) | (36.2–48.5) | (42.5–53.7) | (44.9–55.1) | (35.0–48.8) | (38.4–47.1) |
| **>24 months** | 52.2 | 54.4 | 45.3 | 45.7 | 54.2 | 51.9 |
| | (45.4–59.0) | (48.1–60.5) | (39.4–51.3) | (40.6–50.8) | (47.0–61.2) | (47.4–56.4) |

**Table 4. Wantedness of second adolesecent birth among Uganda women age 20–24 years with first birth <18 years and repeat birth <20, 1995–2016, column % and 95% CI.**

| Survey | 1995 | 2000/01[a] | 2006 | 2011 | 2016 |
|---|---|---|---|---|---|
| Total Sample with repeat births | N = 378 | N = 399 | N = 403 | N = 332 | N = 603 |
| % analyzed for wantedness | 48.7% | 65.7% | 55.8% | 65.1% | 59.7% |
| Then | 75.7 | 72.1 | 58.1 | 61.8 | 54.2 |
| | (67.2–82.6) | (64.9–78.3) | (50.5–65.4) | (52.5–70.3) | (48.3–59.9) |
| Later | 22.5 | 23.1 | 37.6 | 38.0 | 43.1 |
| | (15.8–31.1) | (17.3–30.2) | (30.6–45.1) | (29.5–47.3) | (37.4–49.0) |
| No more | 1.7 | 4.7 | 4.3 | 0.2 | 2.7 |
| | (0.5–5.4) | (2.1–10.5) | (2.1–8.7) | (0.0–1.6) | (1.3–5.7) |

[a]2006; 0.4% missing

We defined repeat adolescent birth as a second or higher live birth following a first birth at <18 years among women age 20–24 years, unlike other studies that used different measurements, thereby making direct comparison of results difficult. For example, a study using the 2011 UDHS defined "rapid repeat pregnancy" as any other pregnancy among married or cohabiting women 16–22 years of age with one or two previous pregnancies, irrespective of

**Table 5. Factors associated with repeat adolescent birth among Uganda women 20–24 years at survey with first birth <18 years, 2016 UDHS (N = 1084).**

| | Total | Repeat adolescent childbirth (row %; 95% CI) | Crude PR (95% CI) | P-value (crude) | Adjusted PR (95% CI) | Wald test P-value |
|---|---|---|---|---|---|---|
| **Residence** | | | | | | |
| Urban | 208 | 44.4 (36.4–52.7) | 1 | | 1 | |
| Rural | 875 | 58.3 (54.5–62.0) | 1.31 (1.08–1.60) | 0.007 | 1.07 (0.89–1.28) | 0.461 |
| **Region** | | | | | | |
| Central | 278 | 45.6 (37.7–53.8) | 1 | | 1 | |
| Eastern | 337 | 63.0 (57.0–68.6) | 1.38 (1.13–1.69) | 0.002 | 1.17 (0.97–1.40) | 0.096 |
| Northern | 216 | 55.4 (49.3–61.4) | 1.21 (0.99–1.50) | 0.069 | 0.97 (0.79–1.19) | 0.760 |
| Western | 252 | 56.9 (50.5–63.1) | 1.25 (1.01–1.54) | 0.040 | 1.10 (0.91–1.34) | 0.328 |
| **Religion** | | | | | | |
| Anglican | 461 | 59.9 (54.6–64.9) | 1 | | 1 | |
| Catholic | 416 | 53.0 (47.5–58.4) | 0.88 (0.78–1.01) | 0.071 | 0.97 (0.86–1.10) | 0.624 |
| Muslim | 172 | 50.3 (41.5–59.1) | 0.84 (0.69–1.02) | 0.075 | 0.97 (0.81–1.15) | 0.696 |
| Other | 036 | 56.2 (39.8–71.4) | 0.94 (0.69–1.28) | 0.687 | 0.98 (0.74–1.29) | 0.865 |
| **Household wealth quintile** | | | | | | |
| poorest | 278 | 67.8 (61.7–73.2) | 1 | | 1 | |
| poorer | 250 | 54.4 (47.7–61.0) | 0.80 (0.69–0.93) | 0.003 | 0.84 (0.73–0.96) | 0.009 |
| middle | 188 | 61.9 (54.3–69.0) | 0.91 (0.79–1.05) | 0.216 | 0.91 (0.78–1.06) | 0.241 |
| richer | 189 | 49.0 (40.2–57.9) | 0.72 (0.59–0.89) | 0.002 | 0.81 (0.67–0.98) | 0.028 |
| richest | 179 | 38.7 (30.1–48.0) | 0.57 (0.45–0.73) | <0.001 | 0.64 (0.48–0.84) | 0.001 |
| **Age at first sex** | | | | | | |
| Mean age(years (SD) | 14.7 (1.42) | 14.3 (1.38) | 0.84 (0.81–0.88) | <0.001 | 1.00 (1.00–1.00) | 0.968 |
| **Age at first birth** | | | | | | |
| Mean age(yrs) (SD) | 15.8 (1.25) | 15.3 (1.30) | 0.76 (0.74–0.79) | <0.001 | 0.77 (0.74–0.80) | <0.001 |

the previous pregnancy outcome [23]. They reported the prevalence of rapid repeat pregnancy of 37% and 74% within 12 and 24 months, respectively. A study from South Africa evaluated prevalence of repeat pregnancies, as an end point, among 13–19 year old black adolescents who were pregnant, recently delivered or had terminated a pregnancy [29]. The authors found a repeat pregnancy prevalence of 17.6% in the first 24 months of observation. Finally, a study in the Philippines using DHS data defined repeated births as an adolescent 15–19 years with atleast two live births. They did not restrict it to those who had first birth <18 years. The authors reported high repeated births among adolescents 15–18 years with negligible reductions over the 20 years with repeated birth declining from 8.49% in 1993 to 7.80% in 2013 [31]. To compare findings on repeat births in adolescence across countries, it could be helpful to decide on a defined measurement that takes into consideration those initiating birth during the most vulnerable period in adolescence, <18 years of age. Births during younger adolescence, <18 years, carries more disadvantages than in older adolescence [8, 36].

Our results suggest that the current generic programs such as those that aim to increase family planning use, have not addressed prevention/delay of repeat adolescent births among adolescents with first birth <18 years. Over time, there was no change in repeat adolescent birth, highest among women from poor households and those reporting first birth at a younger age- each additional yearly decrease in age at first birth was associated with increased likelihood of reporting a repeat adolescent birth. Previous studies demonstrated that the reaction of families and communities in Uganda following first pregnancy/birth might make girls drop out of school, get married, and therefore set on a pathway into repeat adolescent births [11, 12]. Implementation of the law against marriage <18 years act has had challenges with persistence of high sexual and gender-based violence against adolescent girls, including forced marriage/union [45, 46]. This may be sustaining the persistent repeat adolescent births as girls get sent off into marriage at or before first birth and vice versa. The socio-cultural and political positioning in Uganda appears to be perpetuating adolescent births irrespective of the laws and other programs [47]. Our study results show worst statistics during the 2000/01 and 2006 surveys and this was the period when Uganda rollout the implementation of universal primary education and the defilement act (sex or marriage with a girl <18 years was prohibited under the law)- from 1997. The effects of these policies would perhaps have started to be felt after the 2000 survey-in the younger adolescent age groups that had time to benefit from it. Therefore, based on the sample analysed (women 20–24 years), the cohort that would have benefited from these interventions would have been the one surveyed in 2011. Further, contraceptive uptake among sexually active adolescents in Uganda is low, approximately 12%, despite the commitments to improve family planning programs in the country [48–50]. In comparison to factors associated with first adolescent birth in Uganda, after adjusting for age at first birth, age at first sex was no longer significant, but it was significant in crude analysis [38, 39]. Further, repeat birth in adolescence did not appear to be associated with any of the cultural factors such as area of residence or religion that were significant factors for first adolescent birth [38]. This suggests that perhaps it is not so much about when girls start having sex, but rather about contraceptive use and when they have the first birth. A study using the 2011 UDHS investigated rapid repeat pregnancy among Ugandan women 16–22 years [23] and reported that the factors associated with rapid repeat pregnancy among currently married or cohabiting women at interview were: rural residence, region, and age at first union. Our study did not include age at first union in the analysis. Although the greatest burden of repeat adolescent birth (approximately 84%) was among women from rural residence, residence and region were not significant factors at multivariate analysis.

In other LMIC settings, a study in Philippines using DHS data found that repeated pregnancy was more common among adolescents from poorer communities [31]. This study and

another systematic review highlighted the limited information on factors associated with repeat adolescent births in LMICs [21]. Poverty is associated with repeat adolescent pregnancies and births in both LMIC and high-income settings and has a high chance of reverse causality [20, 21]. Poverty perhaps deprives the girl of the power to make decisions over further births, family planning use, access to abortion or by increasing chances of her deciding to complete her family size early due to lack of viable alternatives [20, 21]. Our study further supports that household poverty and young age at first birth appear to be major factors associated with repeat adolescent births.

## Birth interval and wantedness of the repeat adolescent birth

The prevalence of very short birth intervals (<13 months) did not change significantly over the 30 years and remained low 5.4% in 2016, probably due to protection from lactational amenorrhea. Breastfeeding is almost universal in Uganda and mean breastfeeding months being 14.1 months [51]. Over 40% of women reported a birth interval of 13–24 months between the first and second order adolescent births. The failure of the birth spacing of 24 months and less to decline among adolescents with first birth <18 years is in contrast to the decline noted globally among women of reproductive age in LMICs [35]. As fertility has declined in LMICs examined, there has been a lengthening of the birth interval between first and second birth among women 15–49 years. This contrasts with our findings of no significant change in short birth intervals, between first and second birth, among this age category of women. This points to a challenge of family planning uptake among adolescents. Other studies have suggested a phenomenon of the adolescent girls making personal decisions to have more children as a means of building families, reinforcing their motherhood identity, and stabilizing their relationships irrespective of their circumstances and available options [52]. However, our study showed that progressively more women wanted to delay the second order birth despite no reductions in prevalence of repeat adolescent birth over the 30 years. These results support the high unmet need for contraception in Uganda, poor access to reproductive health information, weak adolescents' decision making capacity regarding their reproductive health choices, and short birth intervals, among others [23, 50, 53, 54].

## Limitations

Our study used data from cross sectional surveys that collected self-reported information. This may have been affected by response and recall bias. Further, causality cannot be determined using cross sectional data but rather, associations with the inherent challenge of reverse causality. Further, the study did not explore all possible predictors such as partner-related predictors and influence of previous pregnancy outcome. This study, however, is an important starting point to understand the trends and factors associated with repeat adolescent births using nationally representative samples in sub-Saharan Africa. To reduce on potential recall bias, we analysed data among women age 20–24 years that had a shorter recall period for information pertaining events during their adolescence. This age category had completed the period of observation based on the WHO definition of adolescence period. We do acknowledge that other literature suggests that adolescence may stretch to 24 years of age, with no clear cut off age [55]. In this study, we used live births which does not capture all pregnancies- as others end in abortion or stillbirths. Information on induced abortion in Uganda tends to be under reported as it is prohibited other than for prescribed conditions to save the life of a woman [56, 57]. Contraceptive use was not examined and yet previous studies indicated that it is a predictor of repeat adolescent birth [41].

## Conclusion

Repeat adolescent birth, among women with first birth <18 years, is high in Uganda (more than 1 in 2 women) with no decline observed in 30 years. The mean number of live births by age 20 years among these women did not decline, remaining at 2.2 births. This contrasts with the finding that the percentage having a first birth <18 years has reduced. Birth intervals have not changed over the 30 years, but the wantedness has–women are much less likely to report that they wanted the second adolescent pregnancy at that time. This together might mean that adolescents would like to be supported in avoiding or delaying the second birth, but that the most vulnerable/young/marginalized adolescent girls are being neglected- the poor and those who had first birth at a young age- with services. This is particularly worrying, because adolescents pregnant with their first child would have been very likely to access antenatal, childbirth, postnatal and child vaccination services, which provide ample opportunities to provide postnatal family planning counselling and services. These missed opportunities may keep young mothers in a spiral of repeat adolescent births and we therefore recommend that, every contact between healthcare workers and pregnant adolescents be utilized as an opportunity to prevent unwanted repeat adolescent birth. We suggest future research explores the circumstances and motivators for the repeat adolescent births. This qualitative research should investigate this among women with and without repeat adolescent birth, their parents and partners. There is need to understand how poverty leads to repeat adolescent births and vice versa. Further, understanding of the implementation of the policies and legislation to protect girls <18 years of age is important as a basis for explaining these high repeat adolescent births. Regarding the birth intervals, analysis needs to examine birth interval for first to second childbirth intervals among women 20–24 years with a first birth at or after 18 years so as to determine whether these birth intervals are typical across women who start childbearing both before 18 years and across women who start child birth at and after 18 years.

## Supporting information

**S1 Table. Percent point difference in repeat adolescent birth between surveys among Uganda women age 20–24 years, all UDHS surveys.**
(PDF)

## Acknowledgments

We are thankful to the DHS program and Uganda Bureau of Statistics for permission to use the UDHS data.

## Author Contributions

**Conceptualization:** Dinah Amongin, Annettee Nakimuli, Claudia Hanson, Frank Kaharuza, Lynn Atuyambe, Lenka Benova.

**Data curation:** Dinah Amongin, Mary Nakafeero, Lenka Benova.

**Formal analysis:** Dinah Amongin, Claudia Hanson, Mary Nakafeero, Lenka Benova.

**Funding acquisition:** Dinah Amongin.

**Methodology:** Dinah Amongin, Annettee Nakimuli, Frank Kaharuza, Lynn Atuyambe, Lenka Benova.

**Supervision:** Annettee Nakimuli, Claudia Hanson, Frank Kaharuza, Lynn Atuyambe, Lenka Benova.

**Validation:** Dinah Amongin, Annettee Nakimuli, Claudia Hanson.

**Visualization:** Mary Nakafeero, Lenka Benova.

**Writing – original draft:** Dinah Amongin, Annettee Nakimuli, Claudia Hanson, Mary Nakafeero, Frank Kaharuza, Lynn Atuyambe, Lenka Benova.

**Writing – review & editing:** Dinah Amongin, Annettee Nakimuli, Claudia Hanson, Mary Nakafeero, Frank Kaharuza, Lynn Atuyambe, Lenka Benova.

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
