## [Decision Letter · Decision Letter 0]

3 Feb 2020

PONE-D-19-31884

Time Trends in and factors associated with Repeat Adolescent Birth in Uganda: analysis of six Demographic and Health Surveys

PLOS ONE

Dear Dr Amongin,

Thank you for submitting your manuscript to PLOS ONE. After careful consideration, we feel that it has merit but does not fully meet PLOS ONE’s publication criteria as it currently stands. Therefore, we invite you to submit a revised version of the manuscript that addresses the points raised during the review process.

We would appreciate receiving your revised manuscript by Mar 19 2020 11:59PM. To enhance the reproducibility of your results, we recommend that if applicable you deposit your laboratory protocols in protocols.io, where a protocol can be assigned its own identifier (DOI) such that it can be cited independently in the future. For instructions see: http://journals.plos.org/plosone/s/submission-guidelines#loc-laboratory-protocols

We look forward to receiving your revised manuscript.

Kind regards,

Chaisiri Angkurawaranon

Academic Editor

PLOS ONE

Journal Requirements:

Reviewers' comments:

Reviewer's Responses to Questions

**Comments to the Author**

1. Is the manuscript technically sound, and do the data support the conclusions?

Reviewer #1: Yes

Reviewer #2: Partly

2. Has the statistical analysis been performed appropriately and rigorously? 

Reviewer #1: Yes

Reviewer #2: Yes

3. Have the authors made all data underlying the findings in their manuscript fully available?

Reviewer #1: Yes

Reviewer #2: Yes

4. Is the manuscript presented in an intelligible fashion and written in standard English?

Reviewer #1: Yes

Reviewer #2: Yes

5. Review Comments to the Author

Reviewer #1: This manuscript quantitatively describes repeat adolescent pregnancies in Uganda over almost 30 years. This is a valuable analysis of a meaningful outcome in a vulnerable population, and certainly deserves publication. Please see comments below. Overall, the data (and other previous data) is looked at from many angles and it is important to clarify and highlight the differences in the different statistics offered – eg what is the denominator? is it a rate or a prevalence? etc. I applaud the authors for tackling this complicated subject and look forward to seeing the final publication.

Major Comments:

1. The authors discuss the lack of a standardized definition of repeat adolescent pregnancy as a challenge. However, they chose not to use the definitions previously used in the literature. Please explain the rationale for your definition and explain why the benefit of doing the analysis in this way was big enough to outweigh the drawback of lack of comparability with other reports. (Or re-run the analysis using methods previously described by other authors.)

2. A major advocacy point that emerges from this data is the need for every contact between healthcare workers and pregnant adolescents to be seen as an opportunity to prevent unwanted repeat adolescent birth. This is powerful and actionable. It is amazing that the overall rate of adolescent birth is falling over time, but the rate of repeat adolescent birth (arguably easier to target) is not.

Minor Comments:

1. There are some minor areas of awkward wording in English. Eg:

a. Line 42 and 48: “More women with repeat adolescent births preferred to have…” consider "Many women with repeat adolescent births..." or "An increasing number of ...”

b. Line 68-69: “yet, the way the sexual intercourse came about may be linked to sexual and gender based violence…” consider: “and many of these pregnancies are result of sexual and gender based violence"

c. Line 318-19 “This qualitative research should understand this” consider: “This qualitative research should investigate this” or “This qualitative research should approach this”

2. Line 57-60 wording is unclear. First half of sentence says no change, the second half says there has been a decline. Consider: “These levels have remained high in the last 15 years despite a decline in the age specific fertility rate for Uganda among this category of women, 195 births per 1,000 women in the 1990 Uganda Demographic and Health Survey (UDHS) to 132 births per 1,000 women at the 2016 survey.” Also, I think these numbers should be “per 1,000 women per year”.

3. Line 79: There are other studies that look at adolescent births and then estimate repeat pregnancies (though not the primary outcome) eg: Parker AL, Parker DM, Zan BN, Min AM, Gilder ME, Ringringulu M, et al. Trends and birth outcomes in adolescent refugees and migrants on the Thailand-Myanmar border, 1986-2016: an observational study. Wellcome Open Research. 2018 May 21;3:62.

4. Line 96: How were surveys conducted? How many languages? In person facilitation? This gives an idea of what into the data collection. Could use a reference if it is described elsewhere.

5. Line 123-125: The decisions on these categories seem a little arbitrary but I don't know the religious dynamics in Uganda intimately. I would expect Baptist might be appropriate to group with Anglican and Penticostal under "Protestant"... Alternately, Anglican and Penticostal are quite different in some parts of the world (but maybe not in Uganda). This may be fine as is but it struck me as a bit odd. The “traditional” religion group seems like a potentially interesting cohort but it may have been very small numbers and therefore hard to look at.

6. Line 127-129: these variables that were excluded due to potential reverse causality seem very interesting.

a. Educational level at a certain point predates the pregnancy – for example, “no education” or “some primary school” or “completed primary school” could be interesting variables that would be expected to pre-date the first pregnancy at 14-<18 yo.

b. Occupation of the woman in her early 20’s could be an interesting “outcome” variable, understanding that the association is complex and may be two-way.

7. Methods: Line 159 and onward – please provide the actual numbers in parentheses with the percentages. This helps the reader track the denominators of the different percentages.

8. Table 1: suggest changing “1st birth <18, Repeat” to “1st birth <18, Repeat birth <20” for clarity

9. Table 2: I'm not sure this table adds anything. It is quite busy and a bit confusing. The bottom line is that there is a pretty consistent downward trend in repeat adolescent birth over the time period, but it is not as consistent if you look at just the percentage of women with adolescent birth who go on to have a second adolescent birth - this is an important potential area for intervention. I think this could be more compellingly demonstrated using a graph.

10. Table 3: Again, this is complicated to present. I'm sure number of births was not normally distributed, so numerically presenting mean and SD is not particularly meaningful. Again, graphing means might give a nice visual.

11. Table 5: bottom rows - I think this is not necessary to include... or could go in the supplement. It is obvious that women at the older end of the age range are less likely to have had their children as adolescents within the previous 5 years.

12. Table 6: (and line 267-269) Urban/Rural residence and region both look highly significant in the univariate analysis. Is there a problem with collinearity between these variables that leads to loss of significance in the multivariable analysis? What happens if you include only one (rural/urban or region) in the analysis – do you retain the effect?

13. Discussion: Most of the statistics look the worst during the 2000 and 2006 surveys. Are there any policy or cultural pressures during this time (or before or after) that could explain these changes? It would be interesting to have some context here.

14. Line 257: you talk about the fact that some girls desire this second adolescent pregnancy, but this data shows a large proportion do want to delay. This is an important advocacy point that should be highlighted (and gets a little lost).

15. Line 280-282: Good point. It would be nice to mention breastfeeding rates in Uganda.

16. Line 302-303: abortion (and to a lesser extent the more rare outcome of stillbirth) is an interesting limitation. A comment about whether or not termination of pregnancy (as opposed to spontaneous abortion) is available/legal in Uganda might give some helpful background.

17. Line 319-20: “There is need to understand how poverty leads to repeat adolescent births…” Suggest adding “and vise versa.”

18. Line 322-323: “Early child birth is a trap for many births, and this hasn’t change over time- this needs to be studied more.” This final sentence is a bit abrasive and negative. Other than further study, it doesn’t suggest any action to redeem this problematic situation. I would suggest: "Pregnancy and child birth increase contact with the health care system, and each contact should be seen as an opportunity for education and services to prevent subsequent unwanted pregnancies."

Reviewer #2: 1. Review summary

• This reviewed paper aims to understand trends and factors of repeated births among adolescents in Uganda, where adolescent childbirths are prevalent. The authors’ paper provides a solid contribution to a scant evidence base on repeated adolescent pregnancies in sub-Saharan Africa. As the authors acknowledge, while much has been studied about early childbearing among adolescents, less is known about adolescent childbearing frequency and timing following an initial childbirth.

• For the most part, the arguments framed in the introduction are clear and support the rationale for the methods and analyses undertaken. The claims made in the conclusion are also mostly supported by the data presented in the results. There are however, a number of revisions that could be made to the analysis and framing of the paper, further described below.

2. Overall suggestions for improvements

• There are important revisions that could significantly improve the quality and rigor of the analysis. Broadly these include:

o Focusing the analysis and results on descriptive trends and risk factors of repeated births and pregnancy spacing only (and removing data analysis on pregnancy wantedness). The inclusion of all three outcomes can be confusing and distracting to the reader, and takes away from the overall message and research question focused on childbirth frequency.

o If all three trends analyses are kept, then a clearer connection should be made in the introduction justifying why all three outcomes are being used in the analysis (particular in using ‘pregnancy wantedness’ to show that women often don’t want their second pregnancy).

o Additional and/or clearer information should be provided in the introduction, analysis and discussion about typical birth spacing practices within adolescent and adult populations, and how these spacing practices might affect childbirth repetition among adolescents. For example, it is unclear from this analysis whether repeat pregnancy is in fact a problem in this adolescent population, or whether repeat pregnancy among adolescents is simply reflective of typical birth spacing practices (if two year birth spacing intervals are typical, then it would make sense that a large proportion of women who have a first child birth prior to 17 years will inevitably have another birth prior to 20 years). It appears that the authors attempt to address this by providing additional analyses on birth spacing intervals in the results, but the connection and implications is still unclear for the reader.

o Since this is a trends analysis, the authors should consider including additional risk factor models examining how the different risk factors change (or do not) across different DHS survey rounds (see more specific suggestion below).

3. Suggestions for improvements by section

3.1. Introduction:

• The introduction could be strengthened by including additional information about current birth interval spacing in youth versus those of adult populations, in addition to current literature around pregnancy desires/wantedness (if you keep all three types of analyses in your paper)

• The consequences of repeated births are very well explained in the introduction, but no background information is provided on the risk factors of repeated pregnancies. This information should be included in the introduction to justify your variables selection in the risk factor analysis later on.

3.2. Methods and analysis:

• The rationale for using the DHS 2016 survey round for the risk factor analysis should be explicitly stated (rather than other rounds)

• The risk factor analysis is interesting – if the data are available, you could expand on this analysis by using repeated cross-sectional surveys to assess changing risk factors over time to see how or if predictors have changed over time

• Either in the introduction or methods, you should provide a justification for your choice of variables in the final risk factor model

• Please elaborate what your concerns are about reverse causality (1-2 sentences suffice).

• You may want to explain why other variables widely available in DHS were not included (for example contraceptive use, which you mention frequently in the discussion section). You could also include this in your ‘Limitations’ section.

• The use of the prevalence ratio makes sense and is appropriate. However, the rationale behind using poisson regression is unclear (perhaps 1-2 sentences explaining why poisson, rather than logistic regression is necessary could be helpful to the reader).

3.3. Results

• The contents of Table 1 should be explained prior to the sharing of table results in the text. The text and tables can be confusing to read. For example, in Table 1, it should be clarified that the data in the last row “among women 20-24 years with first birth <18 years” is a sub-set sample of those with a first birth before 18 years (the two rows directly above). This took me a bit to understand and sort out without any explanatory text.

• Along with the above point, a clearer explanation of the contents in Table 2 will help clarify any confusions to the reader. For example, it was unclear what the intervals were for ‘5 years’ versus ‘first 15 years’ versus ‘last 15 years’ versus ’30 years’ in Table 2. Alternatively, you could explore presenting the data in Table 2 in a graph form.

• You could potentially discard the information in Table 3 and in the text related to the mean/median number of births – this does not seem to provide much more valuable information and is not mentioned again in your paper later on.

• The birth interval table (Table 4) could be strengthened by including data for first to second childbirth intervals among women 20-24 years with a first birth at or after 18 years. This could help the reader understand whether these birth intervals are typical across women who start childbearing both before 18 years and across women who start child birth at and after 18 years (However, you may have to expand the sample age range for this additional analysis to include women who are older than 20-24 years).

3.4. Discussion:

• If you keep the ‘ pregnancy wantedness’ analyses, please better elaborate in your discussion how these data help validate your findings on repeated pregnancies among adolescents.

• On lines 238-239, you suggest that a defined measurement might be needed. What definition might you recommend based on the measures you used and analysis you conducted?

3.5. Smaller edits:

• In general, the language is clear. However, there are a number of grammatical errors which should be reviewed and corrected (for example ‘ration’ instead of ‘ratio’)

• The use of ‘wantedness’ can be confusing, perhaps ‘pregnancy desire’ could be used instead (if appropriate)

• Lines 68-72 – the logic of these sentences is not clear

• Line 58 – Recommend including the data for the adolescent fertility rate here (in addition to the childbirth rates you have now)

• Line 105 – I believe you mean “among women 20-24 years old with a first live birth”

• Line 132-133 – Please explain what type of bias this might induce

6. PLOS authors have the option to publish the peer review history of their article (what does this mean?). If published, this will include your full peer review and any attached files.

Reviewer #1: Yes: Mary Ellen Gilder

Reviewer #2: Yes: Esther J. Spindler

---

## [Author Response · Author response to Decision Letter 0]

10 Mar 2020

09th/03/2020

To 

Dr. Chaisiri Angkurawaranon 

Academic Editor

PLOS ONE

Dear Dr. Angkurawaranon,

Re; Response to reviewers’ comments and resubmission of revised manuscript ID PONE-D-19-31884

Time Trends in and factors associated with Repeat Adolescent Birth in Uganda: analysis of six Demographic and Health Surveys

Thank you for reviewing and providing feedback on this manuscript. Please receive the revised copy with specific responses and changes summarized in the table below. 

Reviewers comment Response to comment Line number

Journal Requirements:

 Thank you for this guidance. We have updated the entire manuscript and we believe it meets the journal requirements NA

5. Review Comments to the Author

Reviewer One:

This manuscript quantitatively describes repeat adolescent pregnancies in Uganda over almost 30 years. This is a valuable analysis of a meaningful outcome in a vulnerable population, and certainly deserves publication. Please see comments below. Overall, the data (and other previous data) is looked at from many angles and it is important to clarify and highlight the differences in the different statistics offered – eg what is the denominator? is it a rate or a prevalence? etc. I applaud the authors for tackling this complicated subject and look forward to seeing the final publication.

 Thank you so much for this feedback and comments. We have proceeded to clarify on the different statistics offered, in the details below.

Statistics offered for trends are prevalence and the different denominators are clarified in the methods section for each sub-analysis. Pages 5-8

Major Comments: ONE

1. The authors discuss the lack of a standardized definition of repeat adolescent pregnancy as a challenge. However, they chose not to use the definitions previously used in the literature. Please explain the rationale for your definition and explain why the benefit of doing the analysis in this way was big enough to outweigh the drawback of lack of comparability with other reports. (Or re-run the analysis using methods previously described by other authors.) Thank you so much for this observation. Previous studies examined getting/reporting a repeat adolescent pregnancy as an end point and did not examine pregnancy outcome. This is a challenge as some pregnancies get aborted or end in stillbirths. An outcome of a repeat live birth is of major importance as it leaves the adolescent mother with another offspring to care for.

Younger adolescents, <18 years, face more disadvantages than older adolescents. This group is more vulnerable and a second or higher birth among this category of adolescents, while still in adolescence, may compound the disadvantages for the woman and offspring.

We based repeat birth on live births (rather than pregnancies) because that is what is available from the DHS over the many surveys. Lines 118-120, page 6

Lines 283-287, page 15

Major Comments: TWO 

2. A major advocacy point that emerges from this data is the need for every contact between healthcare workers and pregnant adolescents to be seen as an opportunity to prevent unwanted repeat adolescent birth. This is powerful and actionable. It is amazing that the overall rate of adolescent birth is falling over time, but the rate of repeat adolescent birth (arguably easier to target) is not. Thank you for this very important advocacy point. We have emphasized it in the abstract and conclusion section. Line 48-50, page 3

Line 377-380,

Pages 19 & 20

Minor Comments:

1. There are some minor areas of awkward wording in English. Eg:

a. Line 42 and 48: “More women with repeat adolescent births preferred to have…” consider "Many women with repeat adolescent births..." or "An increasing number of ...”

 Thank you for noting this. We have made the correction. Line 42, page 2

Line 48, page 3

b. Line 68-69: “yet, the way the sexual intercourse came about may be linked to sexual and gender based violence…” consider: “and many of these pregnancies are result of sexual and gender based violence"

 Thank you. This has been corrected. Line 68-69, page 4

c. Line 318-19 “This qualitative research should understand this” consider: “This qualitative research should investigate this” or “This qualitative research should approach this” This adjustment has been made. Line 381 & 382, page 20

2. Line 57-60 wording is unclear. First half of sentence says no change, the second half says there has been a decline. Consider: “These levels have remained high in the last 15 years despite a decline in the age specific fertility rate for Uganda among this category of women, 195 births per 1,000 women in the 1990 Uganda Demographic and Health Survey (UDHS) to 132 births per 1,000 women at the 2016 survey.” Also, I think these numbers should be “per 1,000 women per year”.

 Thank you so much. This has been corrected.

 Line 57-60, 

page 3

3. Line 79: There are other studies that look at adolescent births and then estimate repeat pregnancies (though not the primary outcome) eg: Parker AL, Parker DM, Zan BN, Min AM, Gilder ME, Ringringulu M, et al. Trends and birth outcomes in adolescent refugees and migrants on the Thailand-Myanmar border, 1986-2016: an observational study. Wellcome Open Research. 2018 May 21;3:62.

 Thank you so much for this literature. We have taken this into consideration and cited it. Lines 83-85, page 4

4. Line 96: How were surveys conducted? How many languages? In person facilitation? This gives an idea of what into the data collection. Could use a reference if it is described elsewhere.

 Thank you for noting this. This information has been added Line 108-111,

Page 5 & 6

5. Line 123-125: The decisions on these categories seem a little arbitrary but I don't know the religious dynamics in Uganda intimately. I would expect Baptist might be appropriate to group with Anglican and Penticostal under "Protestant"... Alternately, Anglican and Penticostal are quite different in some parts of the world (but maybe not in Uganda). This may be fine as is but it struck me as a bit odd. The “traditional” religion group seems like a potentially interesting cohort but it may have been very small numbers and therefore hard to look at.

 Thank you for this question. We did this based on the Uganda dynamics for which the minority group were placed under “other”. The Anglican and protestant are similar and are second to Catholics in number. Third is Muslims. We discussed this decision among the 4 Ugandan authors on this study, two of whom are social scientists and provided detailed feedback on the appropriateness of this categorization. NA

6. Line 127-129: these variables that were excluded due to potential reverse causality seem very interesting.

a. Educational level at a certain point predates the pregnancy – for example, “no education” or “some primary school” or “completed primary school” could be interesting variables that would be expected to pre-date the first pregnancy at 14-<18 yo.

b. Occupation of the woman in her early 20’s could be an interesting “outcome” variable, understanding that the association is complex and may be two-way.

 This is a very important observation. Indeed we discussed this issue within the co-author team, and feel we made the best decision given the limitations of cross-sectional survey data. Education level and occupation can be both predictors and outcomes of first birth; we cannot examine such complex relationships from these datasets, unfortunately. Given that we wanted to produce rigorous results and the fact that at the point of survey-when the women were 20-24 years, we thought it would not be rigorous to include them as predictors of repeat adolescent birth (or in fact, confounders for the other variables in the multivariable model), due to potential reverse causality. We agree with the reviewer that it would be great to look at them at the point of second birth- prospectively, in future studies. NA

7. Methods: Line 159 and onward – please provide the actual numbers in parentheses with the percentages. This helps the reader track the denominators of the different percentages.

 Thank you so much. We have included the actual numbers. Lines 182-194, page 9

8. Table 1: suggest changing “1st birth <18, Repeat” to “1st birth <18, Repeat birth <20” for clarity We have made this clarification. Table 1, page 10

Line 195

9. Table 2: I'm not sure this table adds anything. It is quite busy and a bit confusing. The bottom line is that there is a pretty consistent downward trend in repeat adolescent birth over the time period, but it is not as consistent if you look at just the percentage of women with adolescent birth who go on to have a second adolescent birth - this is an important potential area for intervention. I think this could be more compellingly demonstrated using a graph.

 Thank you so much for this guidance. This table summarizes the percent point differences between surveys, the initial 15 years, last 15 years and overall difference. We feel it would be essential for the reader to easily access this information and it has been moved to a supplementary material- S1 Table, attached. 

Further, we have provided a graph for repeat adolescent birth following first birth <18 years and maintained details in table 1, last row. Lines 187 & 188, 

Page 9 

S1 Table- attachment

10. Table 3: Again, this is complicated to present. I'm sure number of births was not normally distributed, so numerically presenting mean and SD is not particularly meaningful. Again, graphing means might give a nice visual.

 We thank the reviewer for this important point. While we agree that a visual might be more aesthetically pleasing, we believe that some of our readers are interested in the precise estimates and confidence intervals (SD, IQR, etc), which is why we would strongly prefer to keep this table as is. 

In terms on the distribution, we present number of births before age 20 among women who are 20-24 years old, so there is not a range of values of parity, like there might be among a population of women in reproductive age. We share the reviewer’s concern about how this distribution is presented, and we therefore show several statistics, including mean, median (among various populations by adolescent fertility pattern), and % with 3+ births. We hope that this captures the breadth of descriptives on this population. NA

11. Table 5: bottom rows - I think this is not necessary to include... or could go in the supplement. It is obvious that women at the older end of the age range are less likely to have had their children as adolescents within the previous 5 years. This guidance is noted. We have deleted those rows and this table is now table 4. Line 228- Table 4,

page 12

12. Table 6: (and line 267-269) Urban/Rural residence and region both look highly significant in the univariate analysis. Is there a problem with collinearity between these variables that leads to loss of significance in the multivariable analysis? What happens if you include only one (rural/urban or region) in the analysis – do you retain the effect? Thank you for noting this. There is no collinearity as each of the five regions has both urban and rural clusters. The variable responsible for the loss of effect is “age at first birth”. NA

13. Discussion: Most of the statistics look the worst during the 2000 and 2006 surveys. Are there any policy or cultural pressures during this time (or before or after) that could explain these changes? It would be interesting to have some context here.

 During this period is when Uganda rollout the implementation of universal primary education and the defilement act (sex or marriage with a girl <18 years was prohibited under the law)- from 1997. The effects of these policies would perhaps have started to be felt after the 2000 survey-in the younger adolescent age groups that had time to benefit from it. Therefore, based on the sample analyzed (women 20-24 years), the cohort that would have benefited from these interventions would have been the one surveyed in 2011. 

We have added this context in the discussion section. Lines 300-306, page 16

14. Line 257: you talk about the fact that some girls desire this second adolescent pregnancy, but this data shows a large proportion do want to delay. This is an important advocacy point that should be highlighted (and gets a little lost). Thank you for bringing this important point to our attention. We have highlighted this advocacy point in the discussion section. Line 341-346, page 18

15. Line 280-282: Good point. It would be nice to mention breastfeeding rates in Uganda.

 Thank you. We have added information on breastfeeding. Line 333 & 334, page 18

16. Line 302-303: abortion (and to a lesser extent the more rare outcome of stillbirth) is an interesting limitation. A comment about whether or not termination of pregnancy (as opposed to spontaneous abortion) is available/legal in Uganda might give some helpful background.

 Thank you. We have added this information. Line 361-364, page 19 

17. Line 319-20: “There is need to understand how poverty leads to repeat adolescent births…” Suggest adding “and vise versa.”

 We have done this. Thank you Line 383, page 20

18. Line 322-323: “Early child birth is a trap for many births, and this hasn’t change over time- this needs to be studied more.” This final sentence is a bit abrasive and negative. Other than further study, it doesn’t suggest any action to redeem this problematic situation. I would suggest: "Pregnancy and child birth increase contact with the health care system, and each contact should be seen as an opportunity for education and services to prevent subsequent unwanted pregnancies."

 Thank you so much for this guidance. We have corrected this. Lines 377-380, page 19 & 20

Reviewer #2:

1. Review summary 

• This reviewed paper aims to understand trends and factors of repeated births among adolescents in Uganda, where adolescent childbirths are prevalent. The authors’ paper provides a solid contribution to a scant evidence base on repeated adolescent pregnancies in sub-Saharan Africa. As the authors acknowledge, while much has been studied about early childbearing among adolescents, less is known about adolescent childbearing frequency and timing following an initial childbirth.

• For the most part, the arguments framed in the introduction are clear and support the rationale for the methods and analyses undertaken. The claims made in the conclusion are also mostly supported by the data presented in the results. There are however, a number of revisions that could be made to the analysis and framing of the paper, further described below. Thank you very much for this feedback and all the suggestions for improvement. We have addressed the revisions as below. NA

2. Overall suggestions for improvements

• There are important revisions that could significantly improve the quality and rigor of the analysis. Broadly these include:

o Focusing the analysis and results on descriptive trends and risk factors of repeated births and pregnancy spacing only (and removing data analysis on pregnancy wantedness). The inclusion of all three outcomes can be confusing and distracting to the reader, and takes away from the overall message and research question focused on childbirth frequency.

 Thank you so much for this important observation and guidance. We analyzed wanteness as part and parcel of birth spacing because it can explain the spacing information obtained. Further, it also speaks to the trends observed. We therefore feel, this information adds to advocacy and therefore programming for this group of women.

However, as highlighted by the reviewer, our most important findings relate to the prevalence of repeat births over time and the predictors, which are the crucial points in our Discussion section. NA

o If all three trends analyses are kept, then a clearer connection should be made in the introduction justifying why all three outcomes are being used in the analysis (particular in using ‘pregnancy wantedness’ to show that women often don’t want their second pregnancy).

 Thank you so much for this. We analyzed wantedness as a descriptor. We have provided more clarification in the introduction on its relatedness to the trends and birth intervals Lines 86-89, page 4 & 5.

o Additional and/or clearer information should be provided in the introduction, analysis and discussion about typical birth spacing practices within adolescent and adult populations, and how these spacing practices might affect childbirth repetition among adolescents. For example, it is unclear from this analysis whether repeat pregnancy is in fact a problem in this adolescent population, or whether repeat pregnancy among adolescents is simply reflective of typical birth spacing practices (if two year birth spacing intervals are typical, then it would make sense that a large proportion of women who have a first child birth prior to 17 years will inevitably have another birth prior to 20 years). It appears that the authors attempt to address this by providing additional analyses on birth spacing intervals in the results, but the connection and implications is still unclear for the reader. Thank you for this helpful feedback. We have provided additional information on birth spacing among adolescents compared to adult women - in the Introduction and Discussion sections. Lines 89-92, page 4 & 5.

Lines 337-340, page 18

o Since this is a trends analysis, the authors should consider including additional risk factor models examining how the different risk factors change (or do not) across different DHS survey rounds (see more specific suggestion below).

 This is a very important recommendation and we are considering this for another paper. We have discussed this in the co-author team and feel that risk factor analyses across surveys do not fit within the objective and scope of this paper. NA

3. Suggestions for improvements by section

3.1. Introduction:

• The introduction could be strengthened by including additional information about current birth interval spacing in youth versus those of adult populations, in addition to current literature around pregnancy desires/wantedness (if you keep all three types of analyses in your paper) Thank you. We have done this Lines 86-92, page 4 and 5.

• The consequences of repeated births are very well explained in the introduction, but no background information is provided on the risk factors of repeated pregnancies. This information should be included in the introduction to justify your variables selection in the risk factor analysis later on.

 We have added this information. Line 72-75, page 4

3.2. Methods and analysis:

• The rationale for using the DHS 2016 survey round for the risk factor analysis should be explicitly stated (rather than other rounds)

 We chose the most recent survey, 2016, for this analysis to tease out the most current risk factors for intervention purposes and to inform policy-makers in Uganda (who have indeed requested this analysis). We have added this rationale. Line 128-130, page 6

• The risk factor analysis is interesting – if the data are available, you could expand on this analysis by using repeated cross-sectional surveys to assess changing risk factors over time to see how or if predictors have changed over time

 Yes, these data are available. We do request that this be analyzed for another paper. NA

• Either in the introduction or methods, you should provide a justification for your choice of variables in the final risk factor model

 We chose variables based on previous literature and on what was available in the DHS data. We have added this justification. Line 146-147, page 7

• Please elaborate what your concerns are about reverse causality (1-2 sentences suffice).

 Thank you. This has been done. Lines 144-146, page 7

• You may want to explain why other variables widely available in DHS were not included (for example contraceptive use, which you mention frequently in the discussion section). You could also include this in your ‘Limitations’ section.

 We have added this explanation in the methods section and limitations section. Briefly, the DHS measure contraceptive use at the time of survey, so this happens after the first and repeat adolescent childbirth and can therefore not be a risk factor. Lines 147-149, page 7

Lines 363-364, page 19

• The use of the prevalence ratio makes sense and is appropriate. However, the rationale behind using poisson regression is unclear (perhaps 1-2 sentences explaining why poisson, rather than logistic regression is necessary could be helpful to the reader).

 Thank you so much for this observation. In scenarios where the likelihood of the outcome is high, above 10%, the modified Poisson regression if preferred to the logistic regression to avoid odds ratios overestimating the prevalence ratios. We did provide this explanation and a reference. Line 162-165, page 8

3.3. Results

• The contents of Table 1 should be explained prior to the sharing of table results in the text. The text and tables can be confusing to read. For example, in Table 1, it should be clarified that the data in the last row “among women 20-24 years with first birth <18 years” is a sub-set sample of those with a first birth before 18 years (the two rows directly above). This took me a bit to understand and sort out without any explanatory text.

 Thank you. This explanation has been provided in text and we have also provided additional detail in the row headings. Lines 180-181, page 9

Table 1

• Along with the above point, a clearer explanation of the contents in Table 2 will help clarify any confusions to the reader. For example, it was unclear what the intervals were for ‘5 years’ versus ‘first 15 years’ versus ‘last 15 years’ versus ’30 years’ in Table 2. Alternatively, you could explore presenting the data in Table 2 in a graph form. Thank you so much for this guidance, we have opted to have this as a supplementary table. This will enable readers easily access the percent point differences between surveys NA

• You could potentially discard the information in Table 3 and in the text related to the mean/median number of births – this does not seem to provide much more valuable information and is not mentioned again in your paper later on.

 Thank you for this suggestion. We feel this provides valuable information on the distribution of the numbers of births, by adolescent fertility pattern. We would be willing to consider placing this table into Supplementary material given editorial guidance. At the moment, we added a few sentences to the text referring to this finding in more detail. NA

• The birth interval table (Table 4) could be strengthened by including data for first to second childbirth intervals among women 20-24 years with a first birth at or after 18 years. This could help the reader understand whether these birth intervals are typical across women who start childbearing both before 18 years and across women who start child birth at and after 18 years (However, you may have to expand the sample age range for this additional analysis to include women who are older than 20-24 years).

 Thank you for this very important observation. We however wanted to restrict our analysis to the trend in birth intervals among this category of women in order to assess if there have been any changes within this category. Expanding it as suggested is a very important recommendation and we have included this in the suggestions for future analysis as it is outside the scope of this paper. Lines 385-389, page 20

3.4. Discussion:

• If you keep the ‘ pregnancy wantedness’ analyses, please better elaborate in your discussion how these data help validate your findings on repeated pregnancies among adolescents. Thank you. We have elaborated this in the discussion section. Line 341-346, page 18

• On lines 238-239, you suggest that a defined measurement might be needed. What definition might you recommend based on the measures you used and analysis you conducted?

 We recommend a definition that looks at those initiating birth <18 years, the most vulnerable group in early adolescence. We have included this information. Line 283-287, page 15

3.5. Smaller edits:

• In general, the language is clear. However, there are a number of grammatical errors which should be reviewed and corrected (for example ‘ration’ instead of ‘ratio’) Thank you so much for this observation. We have corrected the grammatical errors. Lines 244 and 253, page 13

• The use of ‘wantedness’ can be confusing, perhaps ‘pregnancy desire’ could be used instead (if appropriate)

 This word has been used by other publications using DHS data and the DHS also looks at wanting then, later or not all. We opted to retain it as wantedness.

Pregnancy desire might be erroneously interpreted as measured while the woman was pregnant; while on the DHS this is a retrospective question asking women to recall pregnancy wantedness up to 5 years after they had a live birth. NA

• Lines 68-72 – the logic of these sentences is not clear

 These sentences position our age category, having births <18 years, and the fact that they are the most vulnerable to sexual abuse. NA

• Line 58 – Recommend including the data for the adolescent fertility rate here (in addition to the childbirth rates you have now) These were the adolescent fertility rates (15-19 years). We have corrected this. Line 58, page 3

• Line 105 – I believe you mean “among women 20-24 years old with a first live birth”

 Yes, we have corrected this Line 117, page 6

• Line 132-133 – Please explain what type of bias this might induce We have done so Line 153, page 7

---

## [Decision Letter · Decision Letter 1]

26 Mar 2020

Time Trends in and factors associated with Repeat Adolescent Birth in Uganda: analysis of six Demographic and Health Surveys

PONE-D-19-31884R1

Dear Dr. Amongin,

We are pleased to inform you that your manuscript has been judged scientifically suitable for publication and will be formally accepted for publication once it complies with all outstanding technical requirements.

With kind regards,

Chaisiri Angkurawaranon

Academic Editor

PLOS ONE

Additional Editor Comments (optional):

Reviewers' comments:

Reviewer's Responses to Questions

**Comments to the Author**

1. If the authors have adequately addressed your comments raised in a previous round of review and you feel that this manuscript is now acceptable for publication, you may indicate that here to bypass the “Comments to the Author” section, enter your conflict of interest statement in the “Confidential to Editor” section, and submit your "Accept" recommendation.

Reviewer #2: All comments have been addressed

2. Is the manuscript technically sound, and do the data support the conclusions?

Reviewer #2: (No Response)

3. Has the statistical analysis been performed appropriately and rigorously? 

Reviewer #2: (No Response)

4. Have the authors made all data underlying the findings in their manuscript fully available?

Reviewer #2: (No Response)

5. Is the manuscript presented in an intelligible fashion and written in standard English?

Reviewer #2: (No Response)

6. Review Comments to the Author

Reviewer #2: (No Response)

7. PLOS authors have the option to publish the peer review history of their article (what does this mean?). If published, this will include your full peer review and any attached files.

Reviewer #2: Yes: Esther J Spindler

---

## [Editor Report · Acceptance letter]

30 Mar 2020

PONE-D-19-31884R1 

Time Trends in and factors associated with Repeat Adolescent Birth in Uganda: analysis of six Demographic and Health Surveys 

Dear Dr. Amongin:

I am pleased to inform you that your manuscript has been deemed suitable for publication in PLOS ONE. Congratulations! Your manuscript is now with our production department. 

With kind regards,

on behalf of

Dr. Chaisiri Angkurawaranon 

Academic Editor

PLOS ONE